# Genomic Diversity and Antibiotic Resistance of *Escherichia coli* and *Salmonella* from Poultry Farms in Oyo State, Nigeria

**DOI:** 10.3390/microorganisms13061174

**Published:** 2025-05-22

**Authors:** Victoria Olusola Adetunji, Alistair Davies, Tom Chisnall, Mwapu Dika Ndahi, Idowu Oluwabunmi Fagbamila, Eme Ekeng, Ini Adebiyi, Olutayo Israel Falodun, Roderick M. Card

**Affiliations:** 1Department of Veterinary Public Health and Preventive Medicine, Faculty of Veterinary Medicine, University of Ibadan, Ibadan 200005, Nigeria; 2Department of Bacteriology, Animal and Plant Health Agency, Woodham Lane, Addlestone KT15 3NB, UK; alistair.davies@apha.gov.uk (A.D.); thomas.chisnall@apha.gov.uk (T.C.); roderick.card@apha.gov.uk (R.M.C.); 3Department of Veterinary and Pest Control Services, Federal Ministry of Agriculture and Food Security, Area 11, Abuja 900103, Nigeria; whitendahi@yahoo.com; 4National Veterinary Research Institute, Vom 930001, Nigeria; dridowu4u@yahoo.com; 5Nigeria Centre for Disease Control Laboratory, Abuja 900104, Nigeria; eme.ekeng@ncdc.gov.ng; 6University College Hospital, Ibadan 200285, Nigeria; iniadebiyi@yahoo.com; 7Department of Microbiology, University of Ibadan, Ibadan 200005, Nigeria; falod2013@gmail.com

**Keywords:** commercial poultry farms, *Salmonella*, *Escherichia coli*, antimicrobial resistance, Nigeria

## Abstract

Livestock are a source of zoonotic pathogens and bacteria harbouring antimicrobial resistance (AMR) that can pose a threat to public health. This study assessed the burden of antimicrobial resistance (AMR) and genomic diversity of *Salmonella* and *Escherichia coli* at 25 poultry farms in Oyo State, Nigeria. The farm-level occurrence of *Salmonella* was 48%, with 12 serovars identified, including S. Kentucky Sequence Type 198. Notably, 70% of *Salmonella* isolates were resistant to fluoroquinolones, and multidrug resistance was observed in S. Kentucky and S. Derby. The study also found a 52% farm-level occurrence of extended-spectrum beta-lactamase-producing *E. coli*, with four bla_CTX-M_ variants detected (bla_CTX-M-15_, bla_CTX-M-27_, bla_CTX-M-55_, and bla_CTX-M-65_). A genomic analysis revealed the dissemination of bacterial clones between farms, indicating potential routes of transmission. The high occurrence of AMR in both *Salmonella* and *E. coli* presents a potential public health risk, mainly through the consumption of contaminated poultry products. The study highlights the need for improved farm biosecurity and appropriate antimicrobial use to reduce the spread of resistant strains and combat AMR. These findings contribute to the Nigerian National Action Plan for AMR and underscore the importance of ongoing research and interventions in the poultry sector.

## 1. Introduction

Issues relating to antimicrobial resistance (AMR) have for some decades remained a great public health concern. AMR is associated with higher mortality and longer hospitalisations in people [1]. It has been forecasted that by 2050 over 10 million deaths will be attributable to or associated with AMR [1]. The adverse impact of AMR has been predicted to fall most heavily on low- and middle-income countries, such as Nigeria, where the burden of infectious diseases is much higher and healthcare systems less well resourced [2]. AMR also threatens the health of animals and can only be tackled effectively if addressed across human health, animal health, agriculture, and environmental sectors. Indeed, the quadripartite One Health Joint Plan of Action has a dedicated Action Track for AMR, recognising the importance of tackling AMR across these sectors [3]. A significant source of human infectious disease is zoonotic, including through the food chain [4]. Consequently, efforts addressing AMR and antimicrobial use in livestock reared for consumption can benefit animal and public health, the economy and livelihoods, and food safety and security. Nigeria is addressing these challenges through the implementation of a National Action Plan for AMR [5] and a One Health Strategic Plan [6].

Bacterial diseases including non-typhoidal *Salmonella* (NTS) and *E. coli* have been recognised as major foodborne zoonoses globally [4] and in Nigeria [7,8], with poultry and poultry products being a significant source of infection. The poultry industry in Nigeria has experienced remarkable growth in the last few decades and has the second-largest chicken population in Africa after South Africa [9]. Production systems include extensive (backyard production of mainly indigenous breeds), semi-intensive (50–2000 birds) and intensive commercial systems (>2000 improved breed birds) [10]. Commercial farms raise approximately 21% of chickens in Nigeria [10]. The extensive production system comprises almost half of the chicken population and is undertaken by many rural dwellers to provide a source of income, employment and food security [10,11]. The prevalence of non-typhoidal *Salmonella* in Nigerian poultry has been estimated as 21.4%, 43.6% and 47.9% [12,13,14]. Similarly, the prevalence of AMR in *Salmonella* and *E. coli* obtained from Nigerian poultry is high, with multidrug resistance common and a high occurrence of extended-spectrum beta-lactamase (ESBL)-producing *E. coli* [14,15,16,17,18]. There is a high prevalence of ESBL-producing *E. coli* in hospital and community settings in Nigeria, with a 2020 systematic review reporting prevalence rates between 7.5% and 82.3% from 60 studies [19]. Inappropriate use of antimicrobial agents for the prevention and treatment of bacterial infection in poultry is common in Nigeria [20] and may contribute to the development and persistence of AMR in livestock-associated bacteria.

Since the commencement of implementation of the AMR National Action Plan in 2017, Nigeria has been building a One Health AMR surveillance system across the human health, animal health, and environmental health sectors by strengthening institutional capacities, including laboratories. However, significant evidence gaps remain regarding the prevalence and diversity of AMR *Salmonella* and *E. coli* in poultry. This study aimed to address these gaps by sampling for *Salmonella* and *E. coli* at 25 poultry farms in Oyo State in Southwest Nigeria, assessing antimicrobial susceptibility and using whole genome sequencing to define the genetic diversity.

## 2. Materials and Methods

### 2.1. Study Design and Location

A purposive sampling method was used to collect samples from 25 commercial poultry farms (5 farms from each of five Local Government Areas in Oyo State: Ibadan North, Akinyele. Lagelu. Egbeda and Ido). Samples were collected between December 2020 and March 2021. Samples collected included boot swabs for litter (n = 200), waste run-off water (n = 200) and fresh faeces (n = 200). All samples were collected aseptically into sterile sample bags/bottles and immediately transported on ice to the laboratory for storage at 4 °C. The processing of samples commenced within 24 h of collection.

### 2.2. Isolation and Identification of Salmonella Species

Isolation and identification of *Salmonella* was carried out according to ISO 6579-1:2017 guidelines [21]. Boot swabs and environmental swabs were pre-enriched in buffered peptone water (BPW; Oxoid, Hampshire, UK) with a 1:10 dilution, and then incubated at 37 °C for 18–24 h. A 0.1 mL pre-enriched sample was then added separately to three different locations on Modified Semisolid Rappaport Vassiliadis (MSRV; Oxoid, UK) agar medium and incubated at 41.5 °C for 20–24 h. Further, MSRV-positive samples were streaked into xylose lysine deoxycholate (XLD; Oxoid, UK) and incubated overnight at 37 °C [22]. Confirmation was performed by slide-testing isolates using *Salmonella* Poly(O) and Poly(H) antisera (Prolabs).

*Salmonella* isolates were shipped to the UK according to IATA guidelines on charcoal swabs for further testing. The antigenic formula of each strain was then determined using standard methods [23] and serovar assigned according to the White–Kauffmann–Le Minor scheme [24].

### 2.3. Isolation and Identification of Escherichia coli

The isolation of *E. coli* was carried out according to ISO guidelines. The overnight BPW culture was spread onto MacConkey agar (Thermo Scientific, Waltham, MA, USA) and incubated at 37 °C for 18–22 h to isolate the *E. coli* indicator. To screen for ESBL-producing *E. coli*, the overnight BPW culture was also plated onto MacConkey agar and supplemented with 1 mg/L cefotaxime. Lactose-fermenting colonies were subsequently sub-cultured onto Nutrient agar for biochemical testing. Oxidase and indole tests were conducted, with oxidase-negative and indole-positive isolates being confirmed as *E. coli*. The overnight BPW culture was plated out onto MacConkey agar (Thermo Scientific) which was incubated for 18–22 h at 37 °C to obtain indicator *E. coli*. To screen for ESBL-producing *E. coli,* the overnight BPW culture was also plated onto MacConkey agar supplemented with 1 mg/L cefotaxime. Lactose-fermenting colonies were then sub-cultured onto Nutrient agar for biochemical testing. Oxidase and indole testing was carried out with oxidase-negative and indole-positive isolates confirmed as *E. coli*. A preliminary screen of antimicrobial susceptibility was undertaken by disc diffusion for the following antibiotics: ampicillin (10 µg), cefotaxime (30 µg), ceftazidime (30 µg), chloramphenicol (30 µg), gentamicin (10 µg), meropenem (10 µg), pefloxacin (5 µg), tetracycline (30 µg), tigecycline (15 µg) and trimethoprim/sulfamethoxazole (25 µg, 1:19).

A selection of 47 *E. coli* isolates was made based on susceptibility determined by disc diffusion results and included putative ESBL-producers for further testing. The selection was shipped to the UK on charcoal swabs according to IATA guidelines and cultured onto CHROMagar ECC (CHROMagar, Saint-Denis, France) to check for purity. Blue/pale isolates were sub-cultured onto MacConkey (Thermo Scientific) to confirm as lactose-fermenters. Isolates were then sub-cultured onto Nutrient agar (Thermo Scientific) and biochemical tests and repeated for oxidase and indole.

### 2.4. Antimicrobial Susceptibility Testing

Antimicrobial susceptibility testing was conducted using broth microdilution with commercial plates (Sensititre™ EU Surveillance *Salmonella*/*E. coli* EUVSEC1 plate, Thermo Fisher Scientific, 2021), following CLSI and EUCAST guidelines. Each isolate was suspended to a density of 0.5 McFarland in 5 mL of demineralised water. From this suspension, 10 µL was transferred to 11 mL of Mueller Hinton broth to achieve a target inoculum density of 1 × 10^5^ to 1 × 10^6^ CFU/mL. Using a Sensititre AIM, 50 µL of this inoculum was dispensed into each well of the microtitre plate and incubated at 35–37 °C for 18 to 22 h. Fifteen antimicrobials were tested in this manner: amikacin, ampicillin, azithromycin, cefotaxime, ceftazidime, chloramphenicol, ciprofloxacin, colistin, gentamicin, meropenem, nalidixic acid, sulfamethoxazole, tetracycline, tigecycline and trimethoprim. Isolates showing a potential AmpC or ESBL resistance phenotype (i.e., cefotaxime MIC ≥ 0.5 mg/L and/or ceftazidime MIC ≥ 1 mg/L) were further tested using the EUVSEC2 plate, which includes ten antimicrobials, three of which are from the original EUVSEC3 plate (cefepime, cefotaxime, cefotaxime and clavulanic acid, cefoxitin, ceftazidime, ceftazidime and clavulanic acid, ertapenem, imipenem, meropenem and temocillin). *E. coli* NCTC 12241 (ATCC 25922) served as the control strain. Susceptibility was evaluated using EUCAST Epidemiological Cut-off (ECOFF) values or interpretative criteria proposed by the European Food Safety Authority when an ECOFF value was not available [25]. Isolates exhibiting resistance to three or more antimicrobial classes were classified as multidrug-resistant (MDR) [26].

Antimicrobial susceptibility testing was performed by broth microdilution using commercial plates (Sensititre™ EU Surveillance *Salmonella*/*E. coli* EUVSEC1 plate, Thermo Fisher Scientific, 2021), according to CLSI and EUCAST guidelines. A suspension of each isolate was prepared to a density of 0.5 McFarland in 5 mL demineralised water; 10 µL of the suspension was transferred to 11 mL of Mueller Hinton broth to obtain a target inoculum density of between 1 × 10^5^ and 1 × 10^6^ CFU/mL. Fifty microlitres was dispensed into each well of the microtitre plate using a Sensititre AIM and incubated at 35–37 °C for 18 to 22 h. Fifteen antimicrobials were tested in this manner (amikacin, ampicillin, azithromycin, cefotaxime, ceftazidime, chloramphenicol, ciprofloxacin, colistin, gentamicin, meropenem, nalidixic acid, sulfamethoxazole, tetracycline, tigecycline and trimethoprim). Isolates presenting with a putative AmpC or ESBL resistance phenotype (i.e., cefotaxime MIC ≥ 0.5 mg/L and/or ceftazidime MIC ≥ 1 mg/L) were further tested using the commercially available EUVSEC2 plate, which tests 10 antimicrobials including 3 from the original EUVSEC3 plate (cefepime, cefotaxime, cefotaxime and clavulanic acid, cefoxitin, ceftazidime, ceftazidime and clavulanic acid, ertapenem, imipenem, meropenem and temocillin). *E. coli* NCTC 12241 (ATCC 25922) was used as the control strain. Susceptibility was assessed using EUCAST ECOFF values or interpretative criteria proposed by the European Food Safety Authority (EFSA) in the absence of an ECOFF value [23]. Where isolates presented with phenotypic resistance to three or more antimicrobial classes, they were classified as multidrug-resistant (MDR) [26].

### 2.5. Whole Genome Sequencing

DNA extracts were prepared from overnight Luria Broth cultures using the MagMAX™ CORE extraction kit (Thermo Fisher Scientific, Basingstoke, UK) with the semi-automated KingFisher Flex system (Thermo Fisher Scientific, Basingstoke, UK), following the manufacturer’s instructions. The extracted DNA was then processed for whole genome sequencing on the Illumina HiSeq platform. The resulting raw sequences were analysed using the Nullabor 2 pipeline (Seemann et al., 2020) [27], using as reference the published genome *E. coli* K12 (Accession number U00096.2) or *S.* Kentucky strain 201001922 (Accession number CP028357; [28]). Spades was used for genome assembly (version 3.14.1) [29] and Prokka for annotation (version 1.14.60) [27]. The presence of genes and point mutations conferring AMR was assessed using AMRFinderPlus v3.12.8 [30]. The Sequence Type (ST) was determined with MLST (version 2.19.0; https://github.com/tseemann/mlst accessed on 13 April 2025) using the pubMLST database [31]. New STs were assigned in accordance with the procedures of Enterobase [32]. Phylogenetic trees were built with 100 bootstraps using RAX-ML -NG v. 1.0.2 released on 22.02.2021 by The Exelixis Lab [33] and annotated using iTOL version 7.2 [34].

## 3. Results

A collection of 71 isolates obtained at poultry farms in Oyo State, Nigeria, consisting of 25 putative *Salmonella* and 47 *E. coli*, was analysed in this study. This panel comprised all putative *Salmonella* (n = 25), all putative ESBL-producing *E. coli* (n = 18) and indicator *E. coli* (n = 29).

### 3.1. Salmonella Results

Twenty-five putative *Salmonella* isolates were examined, and all were verified as *Salmonella*, except one (which was not examined further). Twelve farms were positive for *Salmonella*, giving a farm prevalence rate of 48% (12/25). Eleven farms were *Salmonella*-positive for boot-swabs (Appendix A). *Salmonella* was detected in water samples at two farms, in feed or feeders at four farms and in one environmental sample (Appendix A). Serotyping revealed 12 different serovars (Table 1). The most commonly detected serovar was *S.* Isangi, followed by *S.* Derby, *S.* Larochelle, *S.* Kentucky and *S.* Dugbe; all these serovars were present at two or more farms (Table 1). All other serovars were detected only once. More than one serovar was detected at five farms (Table 1).

The MIC results for the *Salmonella* isolates are presented in Table 1 and Appendix A. Of the twenty-four *Salmonella*, the two *S.* Dugbe and the five *S.* Isangi isolates were fully susceptible to all the antimicrobials tested with no resistance genes detected. The remaining isolates were all resistant to the quinolones (ciprofloxacin and nalidixic acid). The *S.* Kentucky isolates had high-level resistance to ciprofloxacin (MIC > 4 mg/L) and possessed mutations in the quinolone resistance-determining regions of chromosomal genes *gyrA* and *parC* that are associated with elevated ciprofloxacin MICs [25] (Table 1). In the other isolates, quinolone resistance was mediated by *qnrB* or *qnrS* genes. Multidrug resistance was noted in the three *S.* Derby isolates (ampicillin, gentamicin, quinolones, sulfamethoxazole and tetracycline) and the two *S.* Kentucky isolates (gentamicin, quinolones, sulfamethoxazole and tetracycline), and the corresponding resistance genes are shown in Table 1. The *S.* Elisabethville isolate was resistant to tetracycline and had the *tet*(A) gene. All the *Salmonella* isolates were susceptible to amikacin, azithromycin, cephalosporins (second–fourth generation), colistin, meropenem, tigecycline and trimethoprim (Appendix A). Overall, there was excellent correspondence (98%) between the observed antimicrobial resistance phenotype and the WGS genotype (Table 1 and Appendix A). The *S.* Derby, *S.* Elisabethville, *S.* Kentucky isolates and one *S.* Larochelle isolate harboured additional genes conferring resistance to antimicrobials not present in the EUVSEC3 panel (Appendix A). We noted that the four isolates harbouring *bla*_TEM-215_ also had an IncN plasmid (Appendix A). The Sequence Type was determined using the WGS (Table 1) and, notably, the *S.* Kentucky were ST198.

### 3.2. E. coli Results

Broth microdilution and whole genome sequencing were undertaken on 47 *E. coli* isolates, with MIC results interpreted using EUCAST ECOFF values. Thirty-five indicator *E. coli* were obtained from 21 farms, and of these, six were ESBL-producers (Table 2 and Appendix A). In total, 18 ESBL-producing *E. coli* were obtained from 13 farms, giving an ESBL occurrence rate of 52% (13/25). These isolates were confirmed as ESBL-producers through synergy with cefotaxime and clavulanic acid and showed a corresponding MIC ratio ≥ 8 with cefotaxime following EUCAST guidelines.

There was a high occurrence of resistance (in ≥70% of isolates) to ampicillin, quinolines and tetracycline; 35 isolates (74%) were multidrug-resistant (Table 2 and Appendix A). Resistance to chloramphenicol, gentamicin, sulfamethoxazole and trimethoprim was common (present in 43–66% of isolates). All isolates were susceptible to amikacin, colistin, meropenem and tigecycline. There was excellent correspondence (99%) between the AMR phenotype determined by broth microdilution and the AMR genotype determined by WGS (Appendix A). The AMR genes associated with resistance to the antibiotics tested by broth microdilution are presented in Figure 1.

The ESBL-producing *E. coli* possessed *bla*_CTX-M-15_ (4 isolates), *bla*_CTX-M-27_ (4 isolates), *bla*_CTX-M-55_ (7 isolates), or *bla*_CTX-M-65_ (3 isolates). *bla*_CTX-M-55_, the most detected variant, was present in five different STs and isolated from five farms (Figure 1). High-level ciprofloxacin resistance (MIC ≥ 4 mg/L) was associated with mutations in the quinolone resistance-determining regions of *gyrA* and *parC*, and some isolates additionally had a mutation in *parE* and/or a *qnr* gene (Figure 1; Appendix A). Isolates with ciprofloxacin MIC above the ECOFF value (0.064 mg/L) but below 4 mg/L generally possessed a *qnrS* or *qnrB* gene. Resistance to the remaining antibiotics was correlated with the presence of *bla*_TEM_ (ampicillin); *mph(A)* (azithromycin); *catA1*, *catA2*, *cmlA1*, *catB3* or *floR* (chloramphenicol); *aac(3)-Id*, *aac(3)-IVa*, *aac(3)-IId*, *aac(3)-IIe*, *aac(6′)-Ib4* (gentamicin); *sul* (sulfamethoxazole); *tet(A)*, *tet(B)* (tetracycline); and *dfrA* (trimethoprim) (Figure 1; Appendix A).

There was substantial genetic diversity within the panel of 47 isolates with 32 different Sequence Types (ST) identified, of which three were newly identified in this study (ST17508, ST17509 and ST17510; alleles presented in Appendix A). Five isolates belonged to Clonal Complex 10 (Appendix A). An analysis of the core genome SNP distances identified three pairs of isolates with high sequence identity (<15 SNPs difference; Appendix A), and therefore, the criteria proposed for clones were met [35]. Of these, one pair was from the same farm (VFE0042 and VFE007; ST1585). A second clone was ST206 and comprised isolates VFE059 and VFE058, which had been obtained at different farms and were ESBL-producers harbouring *bla*_CTX-M-27_. The final clonal pair was ST3580 and comprised isolates VFE057 and VFE055. Intriguingly, VFE057 possessed *bla*_CTX-M-55_ whereas VFE055 had *bla*_CTX-M-15_. We noted that although the core genomes were highly conserved the clones differed in AMR gene content (Figure 1), indicating diversity in the accessory genome.

## 4. Discussion

This study assessed the burden of AMR and the genomic diversity of *Salmonella* and *E. coli* in Nigerian poultry farms. The *Salmonella* farm-level occurrence was 48%, similar to previous studies of commercial farms in Nigeria, which have reported a prevalence of 47.9% [12] and 43.6% [13], but higher than 21.4% from a smaller study of ten farms in Oyo state [14]. The twelve serovars identified have been reported previously in poultry in Nigeria [11,12,13], except *S.* Glostrup, which has been identified in Nigerian cattle isolate [36], and *S*. Limete isolated from wastewater in Nsukka, Nigeria [37]. A search of Enterobase [32] (accessed 4 February 2025) using the parameter ‘Sequence Type’ revealed that the specific STs we identified for *S.* Derby, *S.* Dugbe, *S.* Ituri, *S.* Kentucky, *S.* Larochelle and *S.* Limete have been previously described in Nigeria from poultry, livestock or wild animals. However, apart from *S.* Kentucky ST198 and *S.* Larochelle ST22, these STs have very few entries in Enterobase (range 2–53; median 7) suggesting that they have been infrequently described. Similarly, a wide diversity of infrequently reported *Salmonella* serovars has been described for indigenous poultry in North Central Nigeria [11]. Notably, *S.* Enteritidis and *S.* Typhimurium, common agents of salmonellosis worldwide, were not detected in this study. However, this is consistent with previous reports from Nigerian poultry, which describe an absence or low occurrence of these two serovars [11,12,13].

Non-typhoidal *Salmonella* is a significant cause of human foodborne illness and death globally [4], and poultry products are regarded as a major source of infection [38]. In Nigeria, the burden of NTS infections has been estimated as 325,731 cases and a total of 1043 human deaths [7]. Serovars, including *S.* Kentucky and *S.* Larochelle, have been associated with human infection in Nigeria [39]. Furthermore, in this study, 17 isolates (70%) were fluoroquinolone-resistant and therefore met the WHO classification as high-priority pathogens [40]. Due to the wide range of applications and the nature of the diseases treated the WOAH has classified fluoroquinolones as Veterinary Critically Important Antimicrobial Agents [41]. Multidrug resistance (resistance to ≥3 antimicrobial classes), which has been associated with more serious disease in people [42], was present in the *S.* Kentucky and *S.* Derby isolates. The *S.* Kentucky isolates were ST198, a globally distributed MDR lineage isolated from humans, livestock and wildlife [28,43] and constituting an ongoing risk to public health worldwide.

The farm-level occurrence of ESBL-producing *E. coli* was 52%, which compares to Nigerian poultry positivity rates of 32.0% [44], 37.8% [16] and 38.7% [22]. Four *bla*_CTX-M_ variants were present in isolates from this study. In Nigeria, *bla*_CTX-M-55_, the most common variant in this study, has been detected in humans, poultry and cattle [16], and is distributed worldwide, particularly in Asia [45]. The isolates with *bla*_CTX-M-27_ were from four different farms, but all were in ST206, an ST previously associated with this variant in Nigeria [17]. The *bla*_CTX-M-15_ variant has been reported as the dominant type associated with poultry in Nigeria [22], and in this study, it was present in four isolates, each of the different ST from a different farm (Figure 1). Similarly, *bla*_CTX-M-65_ has been reported in Nigerian poultry previously [22] and was present in three STs and three farms in this study (Figure 1). This investigation highlights, together with previous studies, the wide distribution and diversity of ESBL-producing *E. coli* in the Nigerian poultry sector. This diversity is also seen in isolates from hospital and community settings [19], and our findings indicate an ongoing risk to public health through zoonotic transmission of resistant isolates to humans. Enterobacterales resistant to third-generation cephalosporins are classified as critically priority pathogens by WHO [40] and therefore constitute a potential risk to people, that may be exposed through their livelihoods or through consumption of contaminated poultry products. Due to the wide range of applications and the nature of the diseases treated the WOAH has classified third and fourth generation cephalosporins as Veterinary Critically Important Antimicrobial Agents [41]. The role of plasmids in the dissemination of *bla*_CTX-M_ and other AMR genes is widely recognised and has been described for *bla*_CTX-M-15_ in *E. coli* from Nigerian poultry [16,22]. We detected genetic plasmid markers in *E. coli*, including IncN, IncFII, IncH and IncX1; however, a detailed investigation of plasmids and the carriage of AMR genes was beyond the scope of this study.

The high occurrence of resistance to other antimicrobials (including ampicillin, quinolones and tetracycline) and multidrug resistance was notable and consistent with other studies of *E. coli* from poultry in Nigeria [16,17,18,22]. At poultry farms in Oyo State, there is high use of antimicrobial agents such as tetracycline, penicillin and fluoroquinolones and evidence for poor antimicrobial usage practices [20]. Inappropriate antimicrobial use is a major driver for the development of AMR, and the usage patterns in Oyo State may have contributed to the elevated resistance levels observed.

Genomic analysis provided evidence of ESBL-producing *E. coli* clones present at multiple farms. Similarly, five *Salmonella* serovars were present on more than one farm. These results indicate that these bacteria have been disseminated between farms, and future studies could investigate potential routes and epidemiological links, such as day-old chicks, feed, wildlife, farm workers, farm visitors and fomites. Inadequate farm biosecurity practices, as has been found in poultry farms in Oyo State [20], can heighten the risk for the introduction of *Salmonella* and other bacteria onto farms.

In conclusion, this study has provided new insights into the occurrence and diversity of *Salmonella* and *E. coli* in Nigerian poultry. The study has contributed to the Nigerian National Action Plan for AMR, including through the support of AMR research at a Nigerian university. Four different types of *bla*_CTX-M_ genes were identified in *E. coli,* and there was evidence for the dissemination of bacterial clones between farms. Similarly, evidence for the dissemination of *Salmonella* serovars has been presented. Although resistance to quinoline antibiotics was common, isolates were susceptible to other critically important antimicrobials such as meropenem and amikacin. Interventions such as improved farm biosecurity may help to reduce the spread of resistant strains between premises, and appropriate antimicrobial use can contribute to reductions in AMR.

## Figures and Tables

**Figure 1 microorganisms-13-01174-f001:**
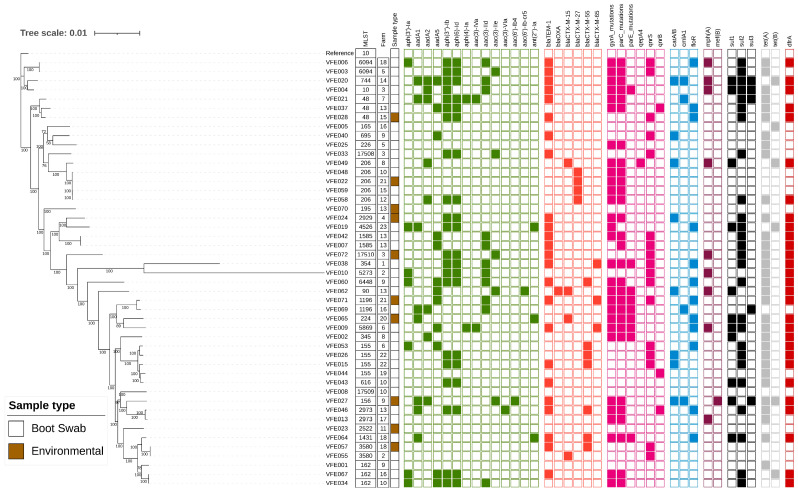
Maximum-likelihood phylogenetic tree generated from core genome single nucleotide polymorphisms of *E. coli* obtained from poultry in Oyo State, Nigeria. The Sequence Type (ST), Sample Type, and anonymised Farm Code are presented. The presence of genes and chromosomal mutations associated with antimicrobial resistance are indicated by filled squares and coloured according to antimicrobial class: green (aminoglycosides), red (beta-lactam), mauve (quinolone), blue (phenicol), purple (macrolide), black (sulphonamide), grey (tetracycline) and brown (trimethoprim). A blank square indicates gene absence. The reference strain was *E. coli* K12 (Accession number U00096.2).

**Table 1 microorganisms-13-01174-t001:** *Salmonella* MIC results. MIC values (mg/L) above the ECOFF value are indicated in bold. Antimicrobial resistance genes and mutations detected shown; a blank cell indicates no corresponding gene/mutation detected. CIP, ciprofloxacin; NAL, nalidixic acid.

Isolate Information				Ampicillin		Quinolones			Gentamicin		Sulfamethoxazole		Tetracycline	
Isolate ID	Farm	Serovar	Sequence Type	MIC	Genotype	CIP MIC	NAL MIC	Genotype	MIC	Genotype	MIC	Genotype	MIC	Genotype
VFS002	20	Derby	9580	**>32**	*bla* _TEM-215_	**1**	**16**	*qnrS13*	**>16**	*aac(3)-IIe*	**>512**	*sul2*	**>32**	*tet(A)*
VFS005	20	Derby	9580	**>32**	*bla* _TEM-215_	**1**	**16**	*qnrS13, qnrB19*	**>16**	*aac(3)-IIe*	**>512**	*sul2*	**>32**	*tet(A)*
VFS008	9	Derby	-	**>32**	*bla* _TEM-215_	**1**	**16**	*qnrS1, qnrB19*	**>16**	*aac(3)-IIe*	**>512**	*sul2*	**>32**	*tet(A)*
VFS009	15	Dugbe	4615	≤1		≤0.015	≤4		≤0.5		64		≤2	
VFS013	10	Dugbe	4615	≤1		≤0.015	≤4		≤0.5		32		≤2	
VFS007	19	Elisabethville	-	≤1	*bla* _TEM-215_	**1**	**16**	*qnrS1, qnrB19*	≤0.5	*aac(3)-IIe*	16	*sul2*	>32	*tet(A)*
VFS011	9	Glostrup	4961	2		**0.5**	**32**	*qnrS1, qnrB19*	≤0.5		128		≤2	
VFS014	22	Isangi	8475	≤1		≤0.015	≤4		≤0.5		64		≤2	
VFS019	22	Isangi	8475	≤1		≤0.015	≤4		≤0.5		32		≤2	
VFS020	3	Isangi	8475	≤1		≤0.015	≤4		≤0.5		64		≤2	
VFS021	6	Isangi	8475	≤1		≤0.015	≤4		≤0.5		64		≤2	
VFS022	22	Isangi	8475	≤1		≤0.015	≤4		≤0.5		64		≤2	
VFS015	20	Ituri	455	≤1		**0.5**	**32**	*qnrB19*	≤0.5		64		≤2	
VFS016	3	Ituri	455	≤1		**0.5**	**32**	*qnrB19*	≤0.5		64		≤2	
VFS024	20	Ituri	455	≤1		**0.5**	**32**	*qnrB19*	≤0.5		256		≤2	
VFS001	12	Kentucky	198	≤1		**>8**	**>64**	*parC*_S80I, *gyrA*_D87G, *gyrA*_S83F	**16**	*aac(3)-Id*	**>512**	*sul1*	**>32**	*tet(A)*
VFS003	6	Kentucky	198	≤1		**>8**	**>64**	*parC*_S80I, *gyrA*_D87G, *gyrA*_S83F	**16**	*aac(3)-Id*	**>512**	*sul1*	**>32**	*tet(A)*
VFS010	5	Labadi	2176	≤1		**0.5**	**32**	*qnrB19*	≤0.5		128		≤2	
VFS006	16	Larochelle	22	≤1		**0.5**	**32**	*qnrB19*	≤0.5		16		≤2	
VFS017	15	Larochelle	22	≤1	*bla* _TEM-215_	**0.5**	**32**	*qnrS13, qnrB19*	≤0.5		32	*sul2*	≤2	
VFS025	15	Larochelle	22	≤1		**0.5**	**32**	*qnrB19*	≤0.5		32		≤2	
VFS018	8	Limete	2617	≤1		**0.5**	**32**	*qnrB19*	≤0.5		128		≤2	
VFS023	15	Nigeria	8405	≤1		**0.5**	**32**	*qnrB19*	≤0.5		32		≤2	
VFS004	3	Telelkebir	2222	2		**0.5**	**16**	*qnrB19*	≤0.5		32		≤2	

**Table 2 microorganisms-13-01174-t002:** Occurrence of resistance to 15 antimicrobials for 47 *E. coli* isolates from poultry in Oyo State Nigeria.

Antimicrobial Class	Antibiotic	Number of Resistant Isolates	% Resistant
Penicillins (aminopenicillins)	Ampicillin	33	70%
Cephalosporins (3rd and 4th generation)	Cefotaxime	18	33%
	Ceftazidime	18	33%
Carbapenems	Meropenem	0	0%
Aminoglycosides	Amikacin	0	0%
	Gentamicin	21	45%
Amphenicols	Chloramphenicol	20	43%
Macrolides	Azithromycin	9	19%
Polymyxins	Colistin	0	0%
Quinolones	Ciprofloxacin	42	89%
	Nalidixic Acid	37	79%
Sulphonamides	Sulfamethoxazole	31	66%
Dihydrofolate reductase inhibitors	Trimethoprim	29	62%
Tetracyclines	Tetracycline	37	79%
Glycylcyclines	Tigecycline	0	0%
Multidrug resistant		35	74%

## Data Availability

The whole genome sequences were deposited in the National Center for Biotechnology Information (NCBI) National Library of Medicine under BioProject accession number PRJNA1246560.

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
