# Peer review of "Genomic Diversity and Antibiotic Resistance of Escherichia coli and Salmonella from Poultry Farms in Oyo State, Nigeria"

_microorganisms, 2025, doi:10.3390/microorganisms13061174_

Round 1
Reviewer 1 Report
Comments and Suggestions for Authors
This work investigate the prevalence of E.coli and Salmonella strains from different settings including various poultry farms in Oyo State, Nigeria, and characterize them with molecular and genomic methods. Due to the current limited such data in Nigeria, the overall findings are interesting and share some insights on the public health issue in AMR. However, the results could be improved to a high-level standard.
- The bacteria names and gene names were not correctly demonstrated in terms of formats. Check them carefully.
- Did authors try to analyze the genetic context of important genes based on sequencing data? Short-read sequencing cannot generate complete genome sequences, so authors may select certain representative strains for nanopore sequencing to do in-depth analysis to make this story more informative.
- Why did authors perform a preliminary screening of AST based on disc diffusion methods? I think authors have conducted broth microdilution to get specific MICs of antimicrobials.
- Did authors compared the prevalence of E. coli and Salmonella in different farms with statistical methods?
- What’s the difference between ESBL-producing E.coli and indicator E.coli strains?
- Plasmids are important facilitators for AMR transmission in E.coli and Salmonella. It's suggested to perform plasmid replicon analysis based on WGS data.
- WGS data was not utilized fully to analyze the AMR genes. Authors could elaborate the E.coli and Salmonella results.
- It’s recommended to analyze the WGS data in this study with human isolates in Nigeria, which will benefit the understanding of One Health concern of such MDR bacteria.
Author Response
Open Review – Review 1
- The bacteria names and gene names were not correctly demonstrated in terms of formats. Check them carefully.
Response: Thank you for spotting this. Bacterial names now all in italics.
- Did authors try to analyze the genetic context of important genes based on sequencing data? Short-read sequencing cannot generate complete genome sequences, so authors may select certain representative strains for nanopore sequencing to do in-depth analysis to make this story more informative.
Response: The authors agree that long-read genome sequencing, such as that provided by nanopore, can provide insights on gene synteny and location that is not readily obtained through short-read methods. However, the use of nanopore sequencing was beyond the scope of the reported work for several reasons, including budget.
- Why did authors perform a preliminary screening of AST based on disc diffusion methods? I think authors have conducted broth microdilution to get specific MICs of antimicrobials.
Response: An initial screen was undertaken by disc diffusion to identify isolates for further analysis by MIC testing and whole genome sequencing. This was to provide value for money as it enabled the costly analysis to be focussed most appropriately.
- Did authors compared the prevalence of E. coli and Salmonella in different farms with statistical methods?
Response: The study was designed to assess farm prevalence rate across the 25 farms examined. The study was not designed to assess prevalence within different compartments at individual farms.
- What’s the difference between ESBL-producing E.coli and indicator E.coli strains?
Response: The effects of consumption of antimicrobials in a given country and animal species, on the occurrence of resistance, can be studied more easily in indicator organisms than in food-borne pathogens, such as Salmonella spp., because all food-producing animals generally carry these indicator bacteria. E. coli is often employed as an indicator organism because it is a common commensal bacteria found in the gut of animals and humans. Furthermore, many laboratories are capacitated to isolate, identify, and characterise E. coli through antimicrobial susceptibility testing. Additionally well-defined interpretive criteria are available for analysis of E. coli susceptibility testing data. In this study (see section 2.3) we screened for the presence of indicator E. coli. We also specifically screened for ESBL-producing E. coli through selective culture in the presence of 1mg/L cefotaxime. We highlight to the reviewer that indicator E. coli can be ESBL-producers.
- Plasmids are important facilitators for AMR transmission in E.coli and Salmonella. It's suggested to perform plasmid replicon analysis based on WGS data.
Response: Plasmids are discussed on e.g. lines 292-296. Plasmid replicon data is presented in Table S1.
- WGS data was not utilized fully to analyze the AMR genes. Authors could elaborate the E. coli and Salmonella results.
Response: The authors believe the WGS data has been analysed to present a detailed insight and contextualisation of the AMR genes detected. It is unclear what additional elaboration the reviewer is referring too.
- It’s recommended to analyze the WGS data in this study with human isolates in Nigeria, which will benefit the understanding of One Health concern of such MDR bacteria.
Response: The manuscript presents a detailed discussion on the importance of the findings in relation to humans in Nigeria. The authors agree that it would be interesting to undertake new and additional analyses along the lines in the reviewer’s comment however these studies are outside the scope of the work presented here.

Reviewer 2 Report
Comments and Suggestions for Authors
The study aims to determine the prevalence and genomic diversity of Salmonella and E. coli​, with a focus on antimicrobial-resistant and ESBL-producing strains, on commercial poultry farms in Oyo State, Nigeria, and to characterize their resistance determinants and patterns of dissemination between farms with WGS.
Since antimicrobial resistance (AMR) in foodborne pathogens is a major concern, especially in low- and middle-income countries where surveillance is limited, and poultry products are a recognized reservoir for zoonotic AMR bacteria, the contribution of this study on farm-level AMR occurrence in a region (Oyo State, Nigeria) is significant.
While previous Nigerian studies have reported prevalence rates of non-typhoidal Salmonella and ESBL-producing E. coli ​(21.4–47.9% for Salmonella and 32.0–38.7% for ESBL-producing E. coli), the identification of 12 serovars including the globally important ST198 S. Kentucky, and the detection of four distinct blaCTX-M variants in diverse E. coli sequence types, are findings that indicate the importance of improved interventions in the poultry industry.
The conclusions are consistent with the results and directly address the research question. The authors conclude that AMR, especially fluoroquinolone and ESBL resistance, is widespread on poultry farms in Oyo State, Nigeria, and that clonal spread occurs between farms, enhancing the existing need for biosecurity and prudent antimicrobial use.
I have only minor suggestions:
- Provide citations for the ISO 6579-1:2017 and ISO 725:2005 standard in the References (currently only laboratory studies are cited as [14, 15]).
- Consider highlighting multidrug-resistant isolates for easier identification in Table 1.
- Consider increasing font size of sample labels in Figure 1.
Overall, I would suggest Accept after Minor Revisions.
Author Response
Open Review – Reviewer 2
- Provide citations for the ISO 6579-1:2017 and ISO 725:2005 standard in the References (currently only laboratory studies are cited as [14, 15]).
- Response: The references has been updated
- Consider highlighting multidrug-resistant isolates for easier identification in Table 1.
Response: Thank you for this suggestion. We considered it but decided not to change Table 1, to keep the presentation of data simple and because the MDR serovars are named in the text.
- Consider increasing font size of sample labels in Figure 1.
Response: The font size for gene names and isolate IDs is fixed to the image size, which we can’t increase larger than a page. The images are high resolution and zoomable and we anticipate that the journal will format the image in the most favourable manner to aid reading.

Reviewer 3 Report
Comments and Suggestions for Authors
The manuscript "Genomic Diversity and Antibiotic Resistance of Escherichia coli and Salmonella from Poultry Farms in Oyo State, Nigeria" deals with a survey and the analysis of antibiotic resistance in Salmonella and E. coli in Nigerian farms. Besides microbiological analyses, the authors carried out antimicrobial susceptibility tests and genomic analyses, which supported the data and provided further insights into the possible resistance mechanisms and diffusion.
Further comments are reported as Notes in the pdf file.

Some improvements are required throughout the text (I've highlighted a few points in the text as notes), though the manuscript can be understood in its present form.
Author Response
Open Review – Reviewer 3
- Please change into Italics and check the same troughout the text
Response: Thank you for spotting this. Bacterial names now all in italics.
- Related
Response:We considered changing ‘relating’ to ‘related’ and on balance determined that the use of ‘relating’ was better suited to this sentence. Therefore, we have not made the suggested change.
- High
Response: The authors agree the figure from the cited reference is high. This helps to illustrate the threat that AMR presents.
- It would be interesting to test the susceptibility to other quinolones, especially those of 4th generation
Response: The authors agree this may be of interest. However, 4th generation quinolones were not present in the commercial plates we employed for MIC testing. The findings as presented for nalidixic acid and ciprofloxacin provide good insight into quinolone resistance and have enabled discussion on the MICs obtained with respect to chromosomal mutations and AMR genes (see e.g. lines 176-180 and 216-220).
- Including some information about possible relationships between the use of specific antibiotics in the region and the occurrence of the observed resistances would be interesting to readers, in my opinion
Response: This is discussed on lines 299-303.

Round 2
Reviewer 1 Report
Comments and Suggestions for Authors
None